# An Analysis of Systems of Influences through the Lens of Balanced Time Perspective: A Qualitative Study on a Group of Inmates

**DOI:** 10.3390/bs13100806

**Published:** 2023-09-28

**Authors:** Rita Zarbo, Andrea Zammitti, Ernesto Lodi, Paola Magnano

**Affiliations:** 1Faculty of Human and Social Sciences, Kore University, Cittadella Universitaria, 94100 Enna, Italy; paola.magnano@unikore.it; 2Department of Philosophy, Sociology, Education and Applied Psychology, University of Padova, 35139 Padova, Italy; andrea.zammitti@unipd.it; 3Department of Humanities and Social Sciences, University of Sassari, Via Roma, 151, 07100 Sassari, Italy; elodi@uniss.it

**Keywords:** systems of influences, time perspective, future, narrative career counseling, prisoners

## Abstract

To respond to the increasing challenges of the XXI century, career guidance is used as a device to reduce inequalities, to expand the range of opportunities for all people, to deconstruct the stereotypes and the stigmatizations that tend to relegate specific social categories to particular working niches, and to offer people the chance to express their differences and diversities. This study reports the results of research aimed at understanding the dynamics of career construction in people with imprisonment experience through the exploration of their systems of influences; the stories of the inmates were collected using My System of Career Influences (MSCI). In the analysis of the narratives and the systems of influences, the focus was placed on the balanced time perspective as a core dimension to foster career construction and to look at future possibilities. The results show that the participants are mainly focused on the past, and their focus on the present is narrow and seems to represent a moment of stalemate, preventing the possibility for inmates to imagine their future. Suggestions for practical implications of career counseling for inmates are provided, and the knowledge about the temporal orientation of prisoners will enable those who do not show any hope of achieving positive interactions to be reached and provide a higher degree of individualization for social rehabilitation proposals.

## 1. Introduction

Career guidance is an important device both for supporting people in building career paths and for reducing inequalities in work inclusion and expanding the range of employment opportunities for all people, including those who are in vulnerable conditions. Work is one of the main individual needs and contributes to people being recognized as active participants in their communities, being productive and, at the same time, build their identities [1]. One of the emerging tasks in career guidance for vulnerable targets is avoiding relegation to specific job niches due to stereotypes and prejudices toward people in specific social categories, and offering them the opportunity to express their differences and diversities. In career counseling interventions, the main challenge to be met is helping people to escape from rigid categorization and labeling, stimulating self-perception based on their potential, resources, and strengths, in order to expand the range of choices that can be perceived as available, reducing the impact of perceived barriers and obstacles [2]. The justice system, as a complex context at risk of perpetuating structural injustice and conditions of vulnerability, is a system in great need of career support intervention. The justice system could represent an environment for vocational and educational guidance that facilitates the expression of career needs where they have less chance of emerging, in persons most at risk of disinvestment from educational paths and work, which might otherwise result in undignified work. Therefore, Robertson [3] underlines that despite there being a critical gap in scientific studies on the effectiveness of inmate career development support interventions, their relevance can surely be argued as they address an unmet career support need by the justice system [4,5].

Although there are still few studies regarding career counseling and career education interventions for offenders, the scientific literature on this issue highlights that there are various risk factors that make prisoners and ex-prisoners particularly vulnerable in job search activity. The following are some examples: the stigma associated with incarceration and a low level of education [6,7,8,9,10]; ex-offenders, compared to other groups with low educational attainment, have lower employment rates [11,12]; the opportunity for employment is generally relegated to specific sectors, such as construction, manufacturing, food service, and retail [13,14]. Furthermore, many inmates do not have a basic education, which is essential for accessing sustainable employment both before and after detention [15], and illiteracy, associated with an almost non-existent professional history, represents an important risk factor in finding a job, as well as being employed [16].

Other research studies have dealt with analyzing the link between professional occupation and the recidivism rate. The results show that recidivism is lower especially among those who find a stable, profitable, and qualitatively good job after their sentence [14,17,18]. Further studies show that people who participate in programs aimed at improving education [15] and job readiness during their period of detention are less likely to re-offend [19,20,21].

Despite the firm recognition of the importance of work and training in the processes of reintegration and in the contrast/prevention of recidivism, the studies in this sector seem to focus exclusively on processes of choice based on the attitudes and occupational options considered by prisoners [22]. Furthermore, “it is necessary not only to offer any employment opportunity but to guide the person towards an informed choice by encouraging the client to take on an active role at the end of the prison sentence” [23] (p. 11); therefore, it is necessary to pay attention to experimentation on career support intervention aimed at promoting social reintegration. Some studies have addressed the evaluation of the effectiveness of counseling interventions carried out with offenders. Moreover, Fitzgerald et al. [24] found that career counseling increases prisoners’ ability to explore and identify career interests, occupational options, job search skills, and goal setting; additionally, Place et al. [25] highlighted that career counseling interventions, through the strengthening of problem-solving and career awareness skills, reduce the probability of recidivism. Furthermore, the importance that work holds in favoring the socio-occupational reintegration of prisoners is widely recognized in the panorama of theories that try to explain the motivations that drive people to commit crimes. According to Rational Choice Theory, for example, having an income from legal work should reduce the likelihood of committing crimes that involve financial gain [26]. Social control theories also explain criminal behavior by referring to the stakes of the criminal act. Holding a job would make criminal behavior less likely because employed people would have too much to lose [27]. Opportunity theories suggest that working individuals spend a lot of time in their workplace and this decreases contact with other activities that may be associated with criminal behavior [28].

Starting from these assumptions, the goal of any intervention in this field should be to promote education and training courses aimed at supporting prisoners and ex-prisoners in planning their own future and in building aspirations, where aspirations have never had voices, due to the rigidity and prejudices of the system and the community. In fact, often, the cause of the failure of life and career trajectories is a failure to overcome stereotypes and structural and cultural barriers; the latter hinder cultural and professional education and training paths and the construction of inclusive, supportive, and sustainable life and career projects [22], which are the fundamental prerequisites for personal autonomy, educational and professional reintegration, and the subsequent abandonment of the circuit of illegality. The real challenge that arises for those involved in career counseling is to facilitate access to work for persons who have finished serving their sentence that is consistent with their aspirations, personal needs, and values and avoiding stigmatizing professional niches into which ex-convicts converge. Contexts and environments can represent risk or protective factors to the extent that they interact with careers in terms of opportunities or barriers, thus assuming different meanings in the self-construction of one’s life trajectories. For this reason, career counseling could contribute to the development of more inclusive contexts because the realization of individual aspirations and dreams (or even just being able to carry out, know, and pursue them) is inextricably linked to the ability of communities to provide places where such dreams and aspirations can take shape.

## 2. Time Perspective in Imprisonment Experience

Time perspective refers to a relatively stable individual mode that tends to take the form of individual differences through which each individual expresses a preference for a temporal focus. The construct was proposed by Zimbardo and colleagues and operationalized as “the way in which individuals, and cultures, partition the flow of human experience into the temporal categories of past, present and future” [29] (p. 1008). In taking any action, individuals rely on memory (past orientation), on expected consequences (future orientation), or directly on experienced stimulation (present orientation). Typically, one of these perspectives overrides the others and significantly influences decisions and the way actions are taken. Individual variability in time perspective depends on the degree of focus on the past, present, or future. The focus on a specific temporal dimension in any individual can be influenced by various cultural, educational, religious, and family factors, and factors associated with the social context of origin [30].

The balanced time perspective (BTP) is a recent elaboration of the construct proposed by Zimbardo, in which individuals are expected to focus on more than one temporal focus, specifically past and future, in a flexible manner. According to Webster [31] it refers to the positive evaluation of both temporal focuses. ‘Balancing’ is defined as the mental ability to flexibly shift between temporal orientations depending on the characteristics of life tasks, situational elements, and personal resources, rather than remaining systematically crystallized in a specific time perspective that appears to be dysfunctional with respect to the situation [32]. Zimbardo himself suggests that the maintenance of adaptive functioning on the physical, mental, and social levels is associated with the balanced time perspective, which flexibly shifts from orientation toward the future to a positive attitude toward the past. BTP appears to play a role in enhancing positive psychological outcomes, such as motivation, subjective well-being, the quality of interpersonal relationships, self-concept, and active involvement in daily life goals [33,34]. Access to positive memories of the past activates positive emotions, increases self-esteem, and supports a sense of purpose and meaning in life [35]. Orientation toward the future, on the other hand, is associated with higher levels of optimism and the anticipation of positive outcomes [36] and strengthens adaptive functioning [37], increasing the likelihood of the success of training programs and coping strategies.

The experience of detention is, per se, strongly linked to the concept of time: the length of imprisonment determined by the sentence is the primary criterion of punishment; as can be guessed, the time spent in prison differs significantly from the time spent in freedom [38]. Moreover, inmates’ perception of time is affected by isolation; they cannot freely organize their activities, and in prison, each day appears identical and is subjected to penitentiary regulations. Isolation may cause a narrowed time perspective [39].

Despite this evidence, it is surprising that very little attention has been paid to the temporal dimensions in the psychological literature on detention and inmates’ re-education [40]. Some scholars [41] examined how prisoners function in isolation based on their temporal orientation. Prisoners focused on the past tend to escape into memories, idealize life before conviction, and ruminate about negative experiences. Present-oriented individuals, especially those with a fatalistic outlook, focus on daily prison life and experience negative emotions such as anxiety, powerlessness, frustration, and hostility. Future-oriented convicts engage in wishful thinking without taking action toward their plans. Therefore, effective future planning after concluding the experience of imprisonment and its effective realization seems impossible or very difficult if the temporal focus is on the past and the future perspective is vague and steeped in fear; similarly, it will not be easy to obtain a prisoner’s willingness to engage in social rehabilitation pursue a path of vocational education in prison if the temporal focus is exclusively on the present. Finally, inmates who keep track of their past experiences run the risk of not learning from their mistakes and not drawing conclusions from previous experiences. Hence, there exists evidence that familiarity with the time perspectives of inmates may prove beneficial in the following ways: firstly, it can aid in targeting those who exhibit a lack of hope towards attaining favorable social interactions; secondly, it can facilitate an elevated level of customization in social re-education initiatives that aim to facilitate job inclusion [41].

## 3. Narrative-Systemic Approach to Career Counseling with Inmates

The socioeconomic characteristics of the world of the third millennium—instability, uncertainty, and unpredictability—require rethinking of the epistemological level first, and then, the theoretical–methodological level of the theoretical frameworks in career construction; the emerging vulnerabilities arising from instability and unpredictability necessarily call to mind the social and inclusive dimensions of career guidance as a device to reduce inequalities, to broaden the range of opportunities for the benefit of all people, to deconstruct the stereotypes and stigmatization that tend to relegate certain social categories to specific job niches, and to provide the opportunity for people to express their own distinctiveness and diversity. Already, in the past decade, career guidance experts have transitioned towards more comprehensive and sophisticated paradigms that recognize the multi-level interplay between individuals, groups, and societal environments in the pursuit of advancing life-career development. These approaches emphasize the importance of social and economic contexts and highlight the need to consider the dynamic interaction between cultural, contextual, and psychosocial factors in the process of life design in order to support career-related human agency among diverse populations [42].

Constructivist (and storied) approaches [43] could be particularly effective and powerful, especially for those clients or in those situations where standardization—of methods, of stimuli, and of assessment—is not useful because it would mean losing relevant information related to uniqueness. Based on qualitative procedures, they increase the potential for comprehending the ‘other’ through the implementation of life-space mapping activities, working actively with clients through the dynamic process of the construction, reconstruction, and co-construction of personal life stories. Of course, the flexibility of methods and tools must be combined with theoretical–methodological rigor on the basis of which the choice of tools and their interpretation are established. The System Theory Framework [44] provides adequate theoretical and methodological support to customize the career counseling process according to the needs of special clients, such as inmates.

Focusing specifically on the role and importance of career counseling for people with prison experience, it is necessary to share some preliminary thoughts: having a criminal record represents a disadvantageous condition in people’s social and work adaptation [45]; people with experience of imprisonment have to deal with social stigma, both implicit and explicit [46], which considerably reduces their opportunities to obtain and keep a job. Stigma encompasses both formal and informal aspects, where the former denotes the restrictive provisions instituted within laws or regulations, while the latter refers to the impact of former incarceration on the evaluation of individuals by employers, landlords, and other relevant parties [47].

Moreover, the motivations that led to criminality can be traced back to histories and experiences that are entirely personal and not generalizable; therefore, they require a highly idiographic approach in their detection and analysis. For these reasons, it is necessary to adopt a methodological model that, starting from the personal story, can allow persons in prison to tell and reconstruct their own personal and career stories, giving space to fears, concerns, barriers—real or perceived—and influences of the past and present, and imagined for the future. From this perspective, qualitative methodology completely changes the roles of the client and the expert within counseling: the former becomes an active teller of their own story; the latter becomes an active and empathic listener. Therefore, narrative approaches recognize the central role of people in being authors of their stories and, thanks to the reflection on them, can offer a space for listening, first, and stimulating a sense of personal agency, later, which is an important goal within career counseling. Building from the narrative themes, the counselor and the client develop new chapter outlines addressing the needs, the emotions, and the influences of the client [48].

## 4. Aim of the Study

The study presented is aimed at enhancing the understanding of the career development of people with detention experiences through the exploration of their cultural and social contexts of origin; specifically, the career development of five individuals with different experiences of detention will be explored through the analysis of their personal and career stories, focusing on the time perspective as a dimension to foster career planning.

## 5. Materials and Methods

### 5.1. Research Design

To address the aims of this study, we implemented a qualitative research design based on the analysis of five interviews conducted using My System of Career Influences (MSCI) [49,50]. This tool has already been used in other qualitative studies, proving to be reliable in the study of career influences [51,52,53,54,55].

### 5.2. Participants and Procedure

The participants were 5 inmates (3 males and 2 females) aged between 20 and 51 years (M = 29.6). All participants are Italians and, at the time of the interview, they must serve less than one year in prison. The characteristics of the participants are summarized in Table 1.

The recruitment procedure of the persons involved in this study began with the presentation of the research project to a prison institution. We contacted three prison institutions in Sicily and two of them confirmed their availability to participate in the study. Each prison’s director authorized the interviews for a limited time; for this reason, it was not possible to involve other candidates in the study.

The stories were collected through individual interviews, using My System of Career Influences (MSCI) [49,50]. The participants that could be involved in the project were indicated by the director of the prison institution, to whom the following inclusion/exclusion criteria were indicated: the participants had to be over 18 years of age, speak Italian fluently, have been in the prison for at least one year, and not be subject to sentences exceeding 3 years. These inclusion/exclusion criteria were chosen because the MSCI proposes reflections on the present, past, and future, and for a person who has to serve a sentence for a long time, it could be complicated to make reflections on their future.

All the interviews were conducted during the year 2022, in person. The interviews lasted between 60 and 110 min. In three cases, based on the requests and needs of the interviewees, it was necessary to split the interview into two different days. The interviews were fully recorded, and then, transcribed, and were carried out by a single researcher, a PhD student at one of the partner universities in the research project. To achieve this, permission from the institutional and individual participants was collected. As interviews can sometimes generate strong emotions, at the end of each interview, the interviewer used part of the time for a debriefing to restore the participant’s emotional state. This was facilitated by the fact that the interviewer had adequate training in psychological counseling. Participants who needed further support were referred to the psychological service within the institution. In no case were the interviews shared with the prison institution; they were only used for the drafting of this article. Participation was voluntary and no economic reward was provided. Participation in the interview was encouraged as it could represent an opportunity for the participants to reflect on their own career plans.

This research was conducted taking into consideration all the guidelines of the code of ethics of the Italian Association of Psychology (AIP) and the Italian Society of Career Guidance (SIO). Furthermore, the research was approved by the internal review board for research in psychology of the Kore University of Enna, with protocol number UKE-IRBPSY-02.22.03.

### 5.3. Instruments

Data were detected using My System of Career Influences (MSCI) [49]. The Italian version of MSCI has been authorized for research aims by T.M. Sgaramella with the permission of the authors. The MSCI is a small booklet consisting of several sections. The first section is entitled ‘My Present Career Situation’ and includes a number of open-ended questions that encourage the participant to reflect on their current professional situation, work experiences, past and current roles, aspirations for the future, decision-making, and support networks.

The second section consists of several diagrams that encourage the client to identify influences. The first diagram is entitled ‘Thinking about who I am’ and identifies intrapersonal influences such as interests, personality, gender, health, and culture. The second system is entitled ‘Thinking about the people around me’ and analyzes social influences (e.g., family or friendship influences). The third system is called ‘Thinking about society and environment’ and takes into account socio-environmental influences, external to the individual, such as public transport.

The third section includes a fourth system entitled ‘Thinking about my past, present, and future, thinking about chance events’. In this system, the participant is invited to make reflections by retracing their past, and then, moving on to the present and the future. This system, including temporal contextualization, was taken into account for the drafting of this paper and subsequent sections.

The subsequent step is the representation of one’s own system of personal influences; it requires the integration of the previous diagrams into one final diagram that brings together the most important influences. After that, the clients are required to translate the graphic representation of their system of personal influences into a narrative form. To do this, the career counselor can use a series of questions as a guide. Finally, through the final task, ‘My action plan’, the participant has the opportunity to identify all the steps needed to implement the actions that will help them build their career.

### 5.4. Data Analysis

In order to analyze the diagrams, we proceeded to conduct thematic coding using NVivo 12 [56]. NVivo 12 is a piece of computer-assisted qualitative data analysis software (CAQDAS) that allows qualitative researchers to sort, organize, and analyze data, with the aim of improving the quality of analysis [57].

With regard to the analysis procedures, two researchers from the research group and the authors of this paper were responsible for conducting the analyses separately. To calculate inter-rater reliability, Cohen’s kappa coefficient was calculated; this index is commonly used in qualitative research to measure inter-rater reliability [58]. Regarding the interpretation of this index, Landis and Koch [59] indicated that values very close to 0 indicate no agreement, values between 0 and 0.20 indicate a light degree of agreement, values between 0.21 and 0.40 indicate a fair degree of agreement, values between 0.41 and 0.60 indicate moderate agreement, values between 0.61 and 0.80 indicate substantial agreement, and values between 0.81 and 1 indicate an almost perfect agreement level. After comparing the initial data, a degree of agreement of 0.89 emerged. This indicates, by multiplying the value by 100, that the initial agreement was 89%. To reach a level of agreement of 100%, the discrepancies were resolved through the intervention of a third researcher, belonging to the research group and an author of this paper. In the end, a version of the analyses was arrived at that satisfied all researchers and the results were processed using this version.

Data analysis started with data coding. Coding means labeling and creating data categories within the data set. This step was facilitated in the case of the MSCI because within the diagram, the participants’ choices were already coded in keywords. Some codes, however, could be superimposed on each other. The advantage of reviewing these encodings using NVivo 12 is that it allows further sorting of the encoded data through the “nodes’’ tool. The software gives the possibility to identify two types of nodes: main nodes and subnodes. In our case, the analysis was conducted taking into account three main nodes: past, present, and future. This allowed us to include subnodes in each node, contextualizing them according to the time perspective, consistent with what is reported in the introduction.

The NVivo analysis allowed us to create a table and graph, which are shown in the Results section. For each subnode, the temporal contextualization was identified and examples from the participants’ narratives were given.

## 6. Results

Table 2 and Figure 1 show the influences on career development that emerged from the different MSCI systems. These were indicated by the participants as “the most important” for each system, and then, reported in the final customized diagram.

With regard to the influences indicated in the ‘Thinking about who I am’ system, they refer to the personality and values of the participants, such as “*I am a sensitive person*”, “*it is important to me to help others*”; the influences that appear in this system are the only ones that occur as constants in all five participants with reference to both the present, the past, and the future.

‘Thinking about the people around me’ is the most ‘crowded’ diagram for the five participants. It consists of both negative past influences, such as “*the lack of guidance from my family*”, “*my ex-girlfriend ruined me*”, and positive ones, such as “*the support of my aunt and uncle*”, and “*my family encouraged me*”. The reported influences also refer to events that negatively marked the relational and affective sphere of the participants, such as “*my father’s imprisonment*”.

In the system ‘Thinking about society and environment’, the most frequent references are related to the lack of job opportunities in the past, such as “*society offered this kind of work*” and “*it was the only chance that presented itself to me*”, and in the future, such as “*coming out of here there will not be many job opportunities for an ex-convict*”.

Finally, in the system ‘Thinking about my past, present and future, thinking about chance events’, the participants refer to negative chance events in the past related to the family sphere, such as “*the separation from my husband*” or “*the bankruptcy of my father’s business*”, in the present to the possibility of staying in the place where they live and concentrating on themselves or to their psychophysical well-being while in prison, and in the future, the reported influences focus more on the possibility of reconciling work and family.

## 7. Discussion

The five individuals interviewed in the prison generally show a narrow, past-oriented time perspective. Descriptions of the future for most of them are limited to a generic outlook and no specific goals emerge from their narratives. The patterns identified suggest a stagnant time perspective with respect to the current situation, i.e., they do not present their time perspective as actively oriented towards building the future but as a persistent consequence of a negative past that seems to influence their vision of the future. The prisoners imagine their lives outside prison, and their expectations of their forthcoming post-incarceration life shall significantly influence their subjective experience of their imprisonment at present [40]. “So, it is not just the past that shapes the present and the future; the (anticipated) future also shapes the present” (p. 172).

Compared to the past, the starting point of all narratives is made to coincide with the events and experiences that triggered the deviant behavior. These events are strongly linked, on the one hand, to a difficult past or dysfunctional family and interpersonal relationships, and on the other hand, to the working sphere, highlighting a limiting socio-economic context, with few viable job opportunities, almost self-prophetic of the deviant career as the only possible alternative. Consistently, there are some prisoners for whom the past is still alive, who have not reconciled with it, often engaging in rumination, and having difficulty in disengaging from memories of traumatic experiences, thus exhibiting negative past orientation [41]. In the narratives of the more recent past, related to the prison experience, the sense of loneliness, feeling guilty for the suffering caused to one’s family, and the fear of abandonment emerge: “*When I was arrested, for the first three months, I forbade my parents and my girlfriend to visit me because seeing their suffering made me feel even worse*”; “*when I was arrested, my thoughts immediately went to my daughters… the fear that they might take the wrong path and be ashamed of me*”. People in prison often experience both a lack of autonomy or privacy and, on the emotional plane, negative emotions such as fear, stress, frustration, anger, boredom, loneliness, sadness, shame, and guilt [60]. The sense of physical and psychological loneliness of prisoner, often linked to the fear of being forgotten [61], derives from the situation of social compression compared to the outside, that forces prisoners to stay away and have limited communication with their loved ones; moreover, it is linked to the difficulty in relating to other people in prison, and from the particular condition in which each person’s behavior, inside the prison, is subject to the constant judgment of both the other prisoners and the prison staff, so that Sykes [62] even stated that “it is not loneliness that afflicts the prisoner, but life in mass” (p. 4).

This is also underlined in the present study, in which the participants seem to focus only on the here and now of the prison situation, report on the difficulty of adapting to the hostile environment of the prison, in which they are surrounded only by other prisoners and are subject to particular rules and hierarchies: “*in prison I don’t identify myself with the other inmates and I don’t have many friends because they are very different from me… […] if you show your hand, that you help the others, quiet, you pass for weak*”, “*This is my first offense, as soon as I came in I was scared, disoriented… I was not accepted by the inmates who saw me as a snobbish girl who wanted to experience the thrill of prison*”. The so-called “prisoners of the moment” are those whose current experience is predominantly characterized by painful encounters associated with serving their sentence; they tend to fixate on the bleakness of their daily routine, the challenges of prison life, and the adverse emotions accompanying it, such as fear, anger, helplessness, and hopelessness, which reflect their present fatalistic orientation [41].

Finally, with respect to the future, the narration of the participants is sparse and generic, and concrete objectives on which to focus are not identified. Several studies have already shown that most of those who leave prison find themselves with no savings, few job prospects [46], and no future plans [63]. “*I would like to reconcile work and family*” or “*I would like to be independent*” are stated without referring to how or through what. The narration is also marked by a lack of motivation and resignation. Expectations of positive reintegration into society do not seem to emerge among the interviewees and the future vision appears to be heavily compromised by the fear of social labeling and prejudice: “*I know, I’m young and I can still do many things, but this won’t happen because I’ve lost the motivation*”; “*The area where I live does not offer job opportunities not only because of the market crisis but also because there aren’t many opportunities for those labeled as offenders*”. Similarly to what is described by Gulla and colleagues [41], the “prisoners of the future” think to the future, but through wishful thinking, escape into unrealistic dreams that, without determined efforts, are destined to fail. For others, a strong fear of the future, anxiety, and uncertainty cause them to avoid thinking about it.

Therefore, focusing on a specific temporal dimension, whether it be past, present, or future orientation, and the individual’s cognitive appraisal of that dimension, whether positive or negative, is likely to have a significant impact on their attitude while incarcerated. This influence may ultimately determine the likelihood of achieving a positive transformation and abstaining from criminal behavior after completion.

## 8. Conclusions and Limitations

This study offers some fruitful insights on vocational guidance and career counseling for individuals with imprisonment experience, thus contributing to an emerging research field that is linked to the overcoming of stigmatization, and the relegation of vulnerable individuals to specific niches.

Addressing the balanced time perspective with adults who experience complex living conditions can help highlight their needs and identify counseling interventions tailored to their specific needs. The way prisoners think about their life outside prison and their expectations of their future life can significantly influence not only their experience of imprisonment but also the outcome of their social and work reintegration by promoting positive visions about the future and curbing the risk of re-offending. Moreover, focusing on the past involves a further risk factor: prisoners’ self-image is often rigidly built around the past (in particular, the crime event) and, in a period of uncertainty and instability in European societies where people are increasingly called to be able to adapt quickly to changes, this crystallization poses additional challenges to career counseling interventions. Therefore, career counseling interventions could support prisoners in working on their future perspectives and on the re-elaboration of their narration both in terms of personal and professional identity. This could also be read in the framework of the theories previously mentioned: work on re-narration would allow for an increase in the probability that people feel able to achieve a legal income (and no longer a criminal one seen as the only possible choice) as predicted by Rational Choice Theory, increasing the chance that they will reach the condition of a more “engaged life” with fewer opportunities to re-experience crime (theories of opportunity) and with a heightened awareness that they have too much to lose (social control theories)

Finally, supporting the career paths of prisoners may have both individual and social impacts on our communities. According to Robertson [3], the interface between career development and criminal justice should not be limited to individual-level interventions to support prisoners, but can also be considered at a community and social level to deliver justice and peace. In this regard, the author concludes that career counseling interventions are useful in improving access to education, training, and employment support, which can improve employment outcomes, e.g., [64,65] reducing recidivism [66] and producing higher levels of professional maturity and self-efficacy [67].

The contributions of this study should be considered while considering its limitations. First, the sample consists of five participants. This size is certainly not representative of all imprisonment histories, especially considering the different crimes perpetrated, the length of stay in prison (our participants stayed in prison for a relatively short-term period), and other demographic and cultural characteristics. Another limitation has to do with the recruitment procedures. In fact, participants were recruited by explaining that it may be helpful to them to think about and analyze their career plans. This could mean that those who participated in the research were more future-oriented; they were probably concerned about their career plans and therefore chose to participate in the research. In conclusion, research and interventions with vulnerable people can take advantage of idiographic approaches that, through narratives, help researchers and practitioners to identify commonalities and differences among people sharing the same experience, drawing useful insights for planning future interventions.

## Figures and Tables

**Figure 1 behavsci-13-00806-f001:**
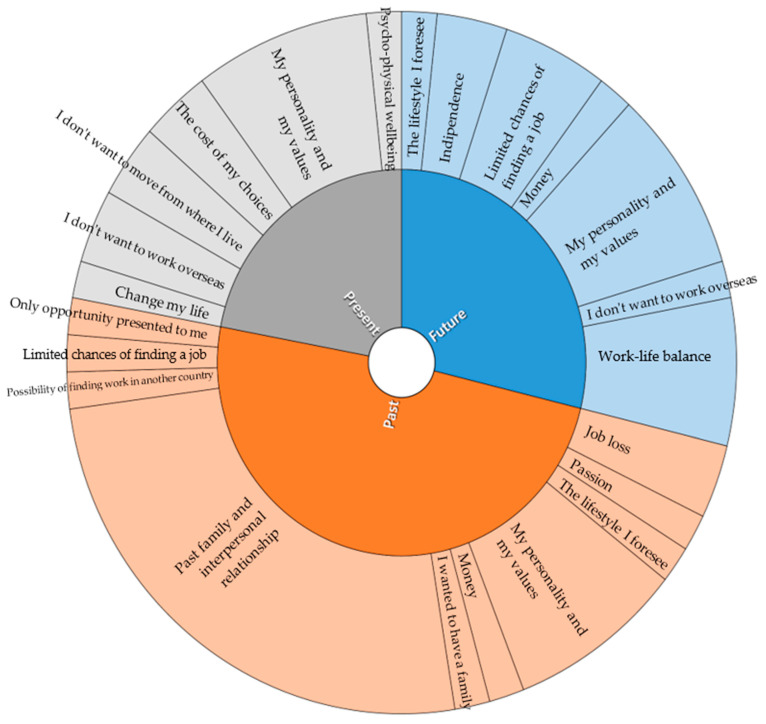
Influences on career development.

**Table 1 behavsci-13-00806-t001:** Demographics.

Name	Age	Gender	Reason for Being in Prison	Year of Arrest
Marco	29	M	Theft	2019
Silvio	20	M	Theft and tax evasion	2019
Alessio	25	M	Bank robbery	2020
Rosa	51	W	Mistreatment of an elderly woman resident in a nursing home	2017
Giulia	23	W	Theft and robbery	2020

Note. M = men; W = women. The names were invented to protect the privacy of the participants.

**Table 2 behavsci-13-00806-t002:** The most important influences.

	Number of Choices	Temporal Contextualization
My personality and my values	5	Past
Past family and interpersonal relationships	15	
Job loss	2	
Passion	1	
The lifestyle I foresee	1	
Money	1	
I wanted to have a family	1	
Possibility of finding work in another country	1	
Limited chances of finding a job	1	
Only opportunity presented to me	1	
My personality and my values	5	Present
Change my life	1	
I don’t want to work overseas	2	
I don’t want to move from where I live	2	
Psycho-physical wellbeing	1	
The cost of my choices	2	
My personality and my values	5	Future
Work–life balance	4	
The lifestyle I foresee	1	
Independence	2	
Money	1	
I don’t want to work overseas	1	
Limited chances of finding a job	3	

## Data Availability

The data are unavailable due to privacy or ethical restrictions.

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
