# Peer review of "An Analysis of Systems of Influences through the Lens of Balanced Time Perspective: A Qualitative Study on a Group of Inmates"

_behavsci, 2023, doi:10.3390/bs13100806_

Round 1

Reviewer 1 Report

The authors interviewed 5 people in prison in Italy, using the My System of Care Influences instrument and semi-structured interviews. They were interested in the way that perspective of time informs how people view the potential for careers, which in turn implies something about the potential for career guidance in prison. 

Overall, I found the topic to be very important and worthwhile. I think the angle of time perspective on career construction is very interesting. My primary concern was that I felt like the data presented was thin. 

INTRODUCTION

Introduction needs to be focused more. It starts by talking about stigmatization and marginalization, but that never comes back meaningfully in the rest of the paper. Instead, the focus of the paper is on career construction and time focus. In general, the flow of the introduction seems loose and should be tightened. 

I also found the first sentence to be confusing. I don't know what it means. 

However, I really liked this section: "the motivations that led to criminality can be traced back to histories and experiences that are entirely personal and not generalizable; therefore, they require a highly idiographic approach in their detection and analysis." I like the perspective that we are not trying to build some society-level anti-crime machine, but to treat people as people. To me, the implication is that a mechanistic approach to crime prevention might be dehumanizing and therefore self-defeating. 

METHODS AND RESULTS

As the authors point out, this is a small group of informants. Given characteristics they list (Italian speaking, at least one year in prison, no more than 3 before release) were there no other candidates in the two prisons? 

I am confused about their methods. In the abstract, they mention they did semi-structured interviews, but the methods just mention the My System of Care Influences. I am not familiar with this instrument, but from the quotes generated, it seems to be a more structured tool. 

I would like to see more quotes to flesh our their Discussion. Overall, this was my biggest issue with the paper. I wanted more from the participants themselves. The results and discussion felt thin, for the importance of the topic. I think this could in turn enrich the final section of the paper. If word count is an issue, I think cutting some parts of the introduction while focusing that section will create space for more in the results and discussion. 

FINAL THOUGHTS

Overall I think this paper was potential. I think it needs more definition in the introduction and more substance in the results and discussion. 

Nothing major. I didn't understand the first sentence, which is a bad start, but the rest was fine. 

Author Response

Dear Reviewer,

Reviewer 2 Report

Summary:

This study focuses on inmates and their time perspective. The authors cite literature on how future time perspective has been linked to career planning and then make a case for why this domain is particularly important for inmates, who may face many barriers to fulfilling careers after serving time. The authors define balanced time perspective (being able to easily switch among past, present, and future orientations) and the benefits of having a balanced time perspective and then carry out qualitative interviews with inmates to assess balanced time perspective. Themes in participants’ responses were identified by two raters and then the themes were coded as past, present, or future temporal orientation. The majority of the themes were past-oriented thinking. The findings are discussed in the context of how understanding time orientation can be useful for career counseling with inmates.

Strengths:

11. Applying future time perspective to this population (inmates) is an interesting idea. Inmates are a vulnerable population when it comes to research and I applaud the authors for their efforts in getting the IRB application approved. I imagine it was a pretty involved process.

22. The authors used an established measure, the My System of Career Influence tool. I feel that this was a more valid and objective approach than unstructured interviews.

Concerns and Areas for Potential Improvement:

11. The authors reference rational choice theory, social control theories, and opportunity theories in their literature review. I think that the paper could be strengthened by linking back to one or more of these theories in the discussion section.

22. The authors state that no incentives were offered for participation and that participants were recruited by explaining that it may be helpful to them to think about and analyze their career plans. This makes me wonder if there was a bias toward the participants being more future-oriented than average, if they were concerned about their career plans while still serving time. It may be worthwhile to acknowledge this as a limitation in the discussion section.

33. The authors state that responses were analyzed and coded by two researchers. Did they calculate interrater reliability?

Author Response

Dear Reviewer,
